# Potential Toxicity of Natural Fibrous Zeolites: In Vitro Study Using Jurkat and HT22 Cell Lines

Michele Betti [1], Maria Gemma Nasoni [1]🄳, Francesca Luchetti [1], Matteo Giordani [2]🄳 and Michele Mattioli [2,*]🄳

1 Department of Biomolecular Sciences, University of Urbino Carlo Bo, 61029 Urbino, Italy; michele.betti@uniurb.it (M.B.); maria.nasoni@uniurb.it (M.G.N.); francesca.luchetti@uniurb.it (F.L.)
2 Department of Pure and Applied Sciences, University of Urbino Carlo Bo, 61029 Urbino, Italy; matteo.giordani@uniurb.it
* Correspondence: michele.mattioli@uniurb.it

**Abstract:** An emerging problem for human health is the exposure to non-regulated mineral fibers with an asbestos-like crystal habit, particularly fibrous zeolites. This study aimed to determine if and how selected fibrous zeolites (erionite, mesolite, and thomsonite) induce toxicity effects on two different in vitro cellular models, the adherent murine hippocampal (HT22) and human immortalized T lymphocyte (Jurkat) cell lines. Before proceeding with the cellular tests, the three zeolite samples were investigated using scanning electron microscopy–energy-dispersive spectroscopy and X-ray powder diffraction techniques. The cells were treated with 0.1 μM and 1 μM of fibrous erionite, mesolite, and thomsonite for 12, 24, and 48 h. Results showed a cytotoxic effect of erionite in both cellular models and revealed different toxic behaviors of the mesolite and thomsonite fibers, suggesting other potential mechanisms of action. The outcome of this study would be a first step for further research on fine biochemical interactions of zeolite fibers with cells and future in vivo investigations.

**Keywords:** erionite; mesolite; thomsonite; fibrous zeolites; cell lines; toxicity





## 1. Introduction

Mineral fibers represent a severe occupational and environmental hazard that may cause interstitial lung fibrosis, pneumoconiosis, and malignant pleural mesothelioma in exposed subjects. The most infamous are the naturally occurring asbestos minerals (chrysotile and five types of asbestiform amphiboles), which are toxic and carcinogenic agents [1–4]. However, other potentially dangerous fibrous minerals have physical and chemical properties similar to asbestos but are not yet regulated. They are identified with the term elongate mineral particles (EMPs [5]) and comprise other fibrous amphiboles (fluoro-edenite, winchite, richterite, glaucophane [6–13]), fibrous antigorite [14], balangeroite [15], and the sulfate epsomite [16].

Recent literature has dedicated particular attention to the natural zeolites with an asbestos-like crystal habit. Fibrous erionite has already been classified as a carcinogenic mineral to humans (group I) by the International Agency for Research on Cancer [17], as confirmed by mineralogical, in vitro and epidemiological studies [18–25]. Following this evidence and in agreement with the remarkable mineralogical similarities, growing interest has concerned other fibrous zeolites such as offretite [26,27], ferrierite [28,29], and mordenite [30,31], thomsonite, and mesolite [32]. However, despite the presence of several works on the mineralogical composition and their physical properties, very few in vitro and in vivo studies on the hazard of these fibrous zeolites have been performed.

Several morphological studies of fibrous zeolites and their interacting ability have been carried out using electron microscopy and electron paramagnetic resonance techniques [21,28,29,31–34]. However, they lack the necessary detail to assess the dependence of the biological activity of zeolite on its microstructural features. Exposure to zeolite fibers such as erionite is related to the formation of reactive oxygen metabolites from

macrophages, induces toxic metal release, alters intracellular homeostasis, and is associated with a necrotic-like effect [35–37]. Several toxicological studies have evidenced that the mechanism by which inhaled fibers induce pathological response is strongly dependent on the diameter, length, aspect ratio, crystal structure, bulk and surface chemical composition, surface reactivity, and biopersistence [13,36,38–43]. Nevertheless, the evidence for the cytotoxicity of zeolites other than erionite in humans is insufficient, and there are very little data from in vivo experiments of the other fibrous zeolites.

To fill this gap, this study investigated the possible toxicity exerted by three selected natural fibrous zeolites: erionite, mesolite, and thomsonite (GF2, GF3a, GF3b respectively). In detail, we performed a comparative in vitro study using human Jurkat and murine hippocampal HT22 cell lines. Our results show the cytotoxic effect of GF2 in both cellular models and reveal different toxic behaviors of the new tested fibers, the mesolite GF3a and thomsonite GF3b, suggesting other potential mechanisms of action. The outcome of this work would be the first step for further research on fine biochemical interactions of zeolite fibers with cells (e.g., intracellular oxidative stress and toxic metal release, DNA damage, inflammatory response).

## 2. Materials and Methods

### 2.1. Natural Zeolite Samples

The GF2 is erionite from the USA (Lander County, NV, USA), which has already been defined as carcinogenic to humans. GF3a and GF3b are fibrous mesolite and thomsonite, respectively, from Iceland (Berufĵorður area, Eastern Region, Iceland). Fibrous mesolite has recently been considered a potentially toxic zeolite, while thomsonite is supposed to be a non-toxic mineral [32].

### 2.2. Scanning Electron Microscopy (SEM)

A scanning electron microscope (FEI Quanta 200 FEG, Hillsboro, OR, USA) was used for morphological investigations and chemical characterization. The operating conditions were: 25 kV accelerating voltage, variable beam diameter, 10–12 mm working distance, and specimen chamber pressure 0.80–0.90 mbar. The images were obtained using a single-shot detector (SSD), while qualitative and semi-quantitative chemical analyses were performed using an energy-dispersive X-ray spectrometer (EDS).

### 2.3. X-ray Powder Diffraction (XRPD)

Pure crystals were selected from each investigated sample and were subsequently pulverized in an agate mortar and loaded in an aluminum sample holder. XRPD data were collected by means of a Philips X'Change PW1830 powder diffractometer (Philips X'PERT, Malvern Panalytical, Almelo, The Netherlands) using the following analytical conditions: 35 kV accelerating voltage, 30 mA beam current, CuK$\alpha$ radiation ($\lambda$ = 1.54506 Å), 2 to 65° 2$\theta$, step size 0.01° 2$\theta$, and 2.5 s counting time. X'Pert Quantify software (Philips X'PERT, Malvern Panalytical, Almelo, The Netherlands) was used for data collection and instrument control, and X'Pert HighScore Plus for semi-quantitative phase analysis. An internal calibration standard (quartz) and a set of three repeated measurements were carried out for each sample.

### 2.4. Cell Culture and Treatments

Murine hippocampal cells HT22 were kindly provided by Professor Herrera Federico, University of Oviedo, Spain. HT22 were cultured in DMEM-HAM'S F12, supplemented with 10% fetal calf serum, L-glutamine (100 mM), and 1% antibiotics (penicillin, streptomycin) (https://onlinelibrary.wiley.com/doi/10.1111/jpi.12747) (accessed on 1 January 2019). The Human Jurkat cell line was provided by ATCC. The cells were cultured in RPMI medium supplemented with 10% fetal calf serum, L-glutamine (100 mM), and 1% antibiotics (penicillin, streptomycin). Cells were incubated in a humidified 5% $CO_2$ atmosphere at 37 °C. The fibers were resuspended in physiological solution, sonicated for 45 s at 100 W



and immediately incubated with the cells. The cells were treated with 0.1 μM and 1 μM of gently powdered fibrous zeolites (GF2, GF3a, GF3b) for 12, 24, 48 h.

### 2.5. Trypan Blue Test

Jurkat cells were seeded at $2 \times 10^5$ cells/mL in 24-multiwell dishes. The treated cells were collected and centrifuged at $300 \times g$ for 5 min. Cells were then resuspended in an equivalent volume of 0.4% Trypan Blue solution and counted with a Burker's chamber under light microscopy. Cells excluding Trypan blue were considered viable.

### 2.6. Clonogenic Test

To explore the potential cytotoxic effects of zeolites, we investigated the clonogenic survival of HT22 cells by counting the number of colonies. $5 \times 10^2$ cells/mL HT22 cells were seeded in six-multiwell dishes and incubated at 37 °C to allow their adhesion to the dish. Upon adherence, the cell culture medium was refreshed (2 mL/well), and cells were subsequently treated at the indicated doses for 12, 24, or 48 h. After the incubation time, the treatment was removed and the cells were then incubated at 37 °C for seven days. The time has been chosen on the basis of HT22 cell proliferation. Cells were fixed and stained using a solution of 50% ethanol and 0.25% methylene blue to detect colonies. Colonies containing at least 50 cells were counted under the microscope. The bright-field microscopy images were acquired by optical inverted microscope using the 10× objective with data acquisition software (Nikon ECLIPSE TS100, software NIS-Elements F, Nikon Europe BV, Amstelveen, The Netherlands).

### 2.7. Statistical Analysis

Data are shown as mean ± standard deviation (SD) of at least three independent experiments. Analysis of variance (ANOVA) approaches was used to compare values among different experimental groups. Differences between groups were analyzed using a Two-way ANOVA analysis of variance, followed by a Bonferroni post hoc analysis. *p* values less than 0.05 were considered statistically significant. All statistical analysis was done using GraphPad Prism 5.0 (GraphPad software, 2365 Northside Dr. Suite 560 San Diego, CA 92108 USA).

## 3. Results

The GF2 sample is an extremely fibrous erionite with a typical wooly aspect (Figure 1A), very similar to the samples observed by Mattioli et al. [21] and Staples and Gard [44]. The fibers are grouped in curved bundles that tend to separate into small fibrils, ranging from 10–20 nm to 0.2–0.3 μm in width. The chemical composition is dominated by potassium and calcium as extra-framework cations (Figure 1A), agreeing with its crystal-chemical formula ($Ca_{2.03}Na_{0.73}K_{2.52}Mg_{0.26}Al_{8.22}Si_{27.78}O_{71.80}$ $35.95H_2O$; [45]). The composition is consistent with Mattioli et al. [21] and Gude and Sheppard [46], allowing its classification as erionite-K.

The GF3a mesolite sample consists of tiny fibers aggregated together, forming a chaotic and intertwined framework (Figure 1B). Most of the fibrils (about 90%) have a diameter equal to or less than 0.5 μm, according to the detailed morphometric data from Giordani et al. (2022). Using the equation of Gonda and Abd El Khalik [47], Giordani et al. [32] calculate an equivalent aerodynamic diameter ($D_{ae}$) of 0.82 μm for this sample. Consequently, the mesolite fibers of the GF3a can easily penetrate the respiratory tract. Microchemical analyses of GF3a crystals show a composition dominated by calcium and sodium as extra-framework cations (Figure 1B), in agreement with the crystal-chemical formula ($Na_{15.7}Ca_{16.4}Al_{46.9}Si_{73.1}O_{240.7}$ $64H_2O$ [32]).

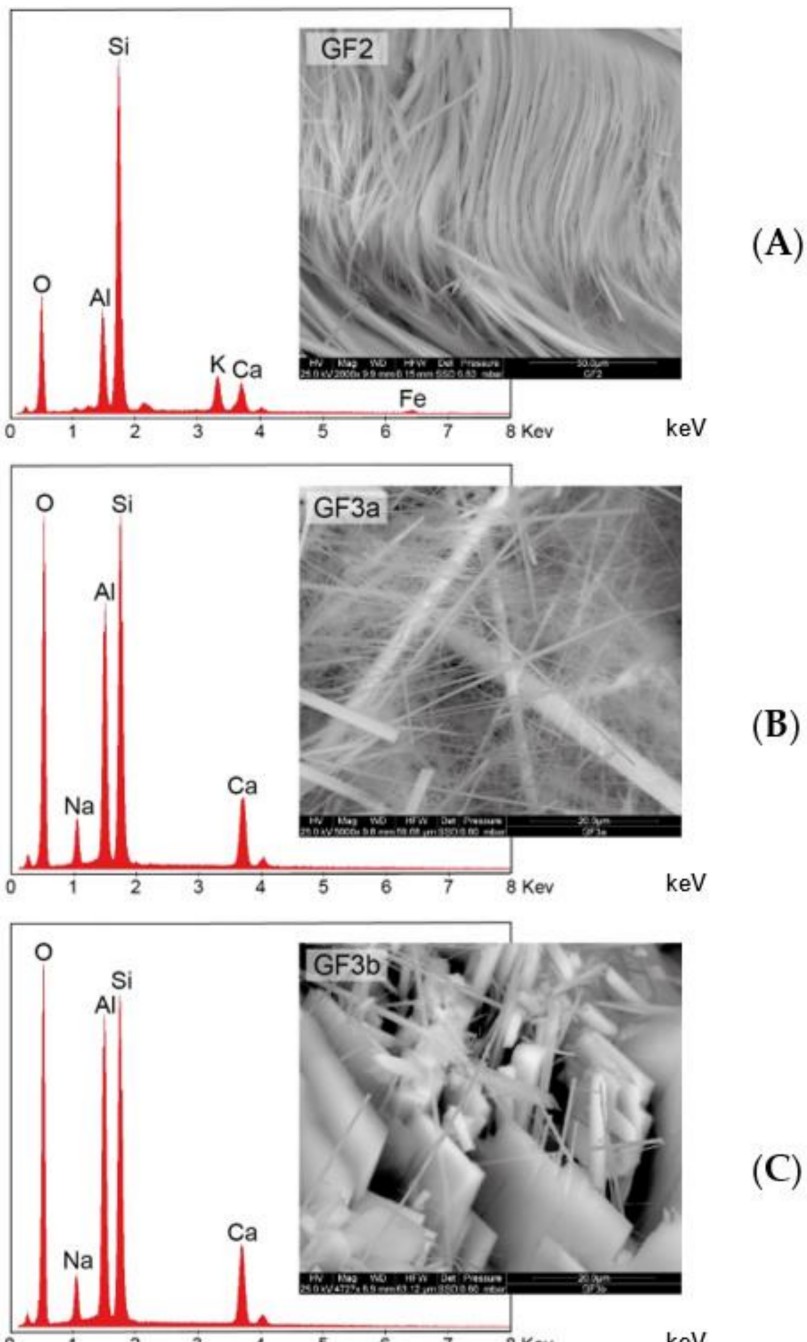

**Figure 1.** SEM images and representative EDS spectra of the investigated zeolite fibers: (**A**) erionite GF2, (**B**) mesolite GF3a, and (**C**) thomsonite GF3b.

The GF3b thomsonite is formed by prismatic crystals and lamellae of millimetric size (Figure 1C). The fragments and crystals rarely are <3 μm, ranging from 1–5 μm to 60 μm. The chemical composition is very similar to that of the GF3a sample, being dominated by calcium and sodium extra-framework cations (Figure 1C). It also agrees with its calculated crystal-chemical formula ($Na_{5.4}Ca_{6.6}Al_{18.0}Si_{22.0}O_{80.3}$ 24$H_2O$ [32]).

To confirm their mineralogical composition and exclude the presence of impurities, crystals from each investigated sample were analyzed by X-ray powder diffraction. Experimental XRPD patterns are displayed in Figure 2, confirming the purity of the investigated samples.

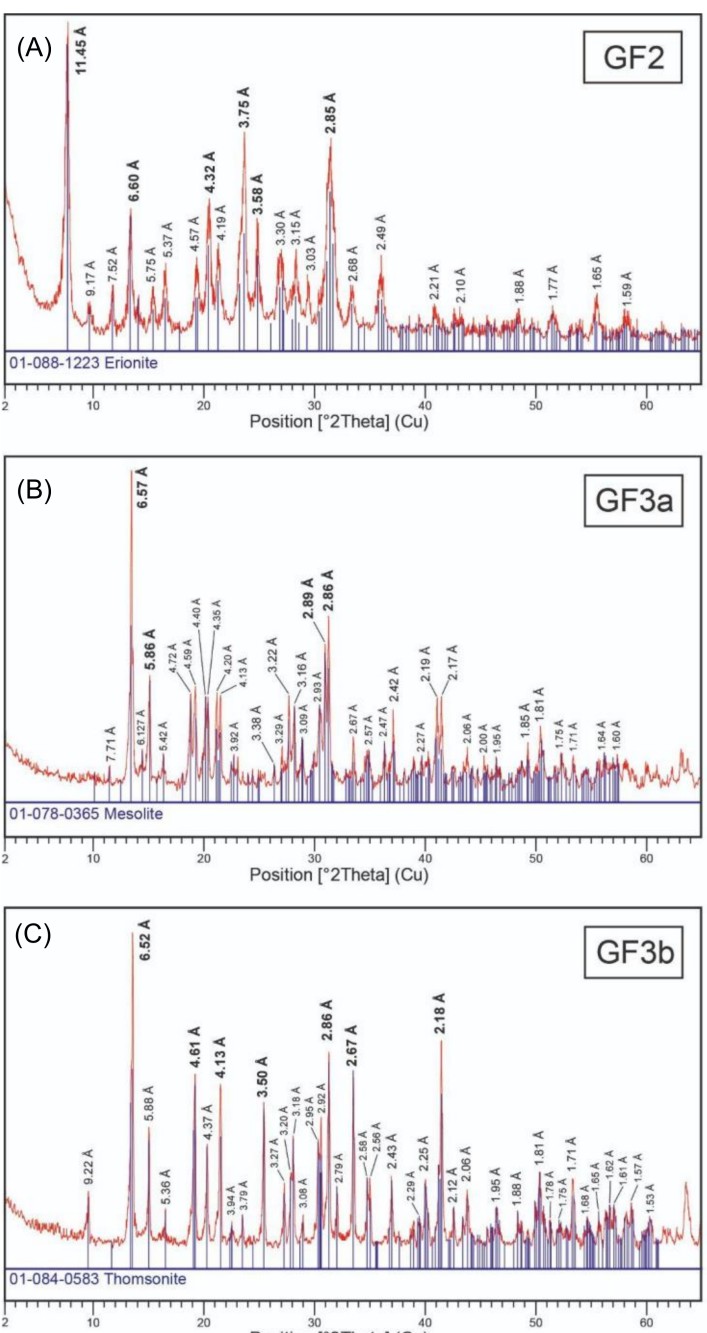

**Figure 2.** XRPD patterns of the investigated samples. The main diagnostic peaks are in bold. (**A**) erionite GF2, (**B**) mesolite GF3a, and (**C**) thomsonite GF3b.

The GF2 sample shows the typical pattern of pure K-erionite. The highest peak intensity (100) occurred at 2θ ~7.7°, corresponding to a d-value of 11.45 Å. Two secondary reflections were at 3.75 Å and 2.85 Å, respectively. Other significant peaks were set at values of d-spacing of 6.60 Å, 4.32 Å, and 3.58 Å.

In the GF3a sample, a pure mesolite composition was well recognizable because it had a high-intensity peak at 2θ = 13.5, corresponding to a d-value of 6.57 Å, accompanied by further notable reflections at 5.86, 2.89, and 2.86 Å. The typical dense association of peaks ~19–22 2θ (i.e., from 4–72 to 4.13 Å) is another characteristic feature of this mineral phase.

The GF3b sample was pure thomsonite and showed the main reflection at 2θ = 13.5 (6.52 Å); other diagnostic characteristic peaks corresponded at d-values of 4.61, 4.13, 3.50, 2.86, 2.67, and 2.18 Å.

To evaluate the potential cytotoxicity of the fibrous zeolites GF2, GF3a, and GF3b, we performed Trypan blue assay in Jurkat cells and a clonogenic test in adherent murine hippocampal HT22 cell line after 12, 24, and 48 h of treatment. Initially, three doses (0.1, 1, and 10 μm) were considered for investigating toxic effects since the highest concentration showed lethality above 90% at 12 h data are not shown.

The results depicted in Figure 3A show a dose and time-dependent decrease of cell viability in the HT22 cell line for both concentrations tested (0.1 and 1 μm). In detail, we observed a significant effect of GF3a on cell viability within 12h of exposure with the lower concentration (* $p < 0.05$). At the same time, the fibrous zeolites GF2 and GF3b do not show a cytotoxic effect. On the other hand, the cytotoxic effect is evident with the higher concentration after 12, 24, and 48 h of exposure. In addition, the representative images of the clonogenic assay (Figure 3B) also highlighted a dose-dependent reduction of colony formation after the treatment with all three fibrous zeolites tested.

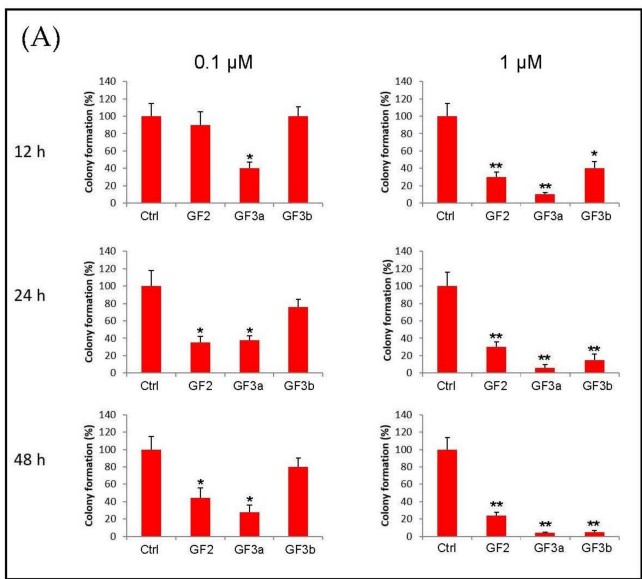

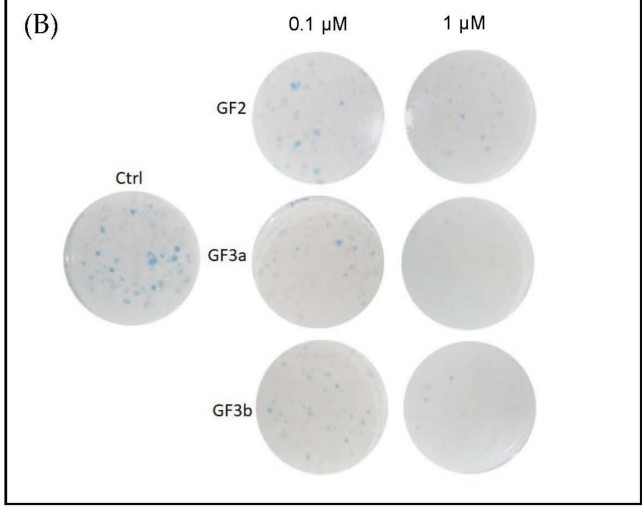

**Figure 3.** Evaluation of potential toxicity of GF2, GF3a, GF3b fibrous zeolites on HT22 cell line. (**A**) The graphs depict the percentage of colony formation after 12, 24, and 48 h of treatment with the three fibrous zeolites GF2, GF3a, and GF3b at the concentration of 0.1 μM and 1 μM. Each value is expressed as a percentage $\pm$ SD (N = 3 independent experiments performed in triplicate; * $p < 0.05$, ** $p < 0.01$ vs. Ctrl). (**B**) Representative colony images of HT22 cells treated with GF2, GF3a, and GF3b fibrous zeolites after 24 h of treatment.

The representative bright-field microscopy images in Figure 4 show the cytotoxicity of the higher concentration GF2, GF3a, and GF3b fibrous zeolites on HT22 cells. In detail, the HT22-treated cells lost their original flat and elongated morphological features. In addition, as revealed by the enlarged inserts, the cytoplasm appeared more expanded with a higher presence of vacuoles.

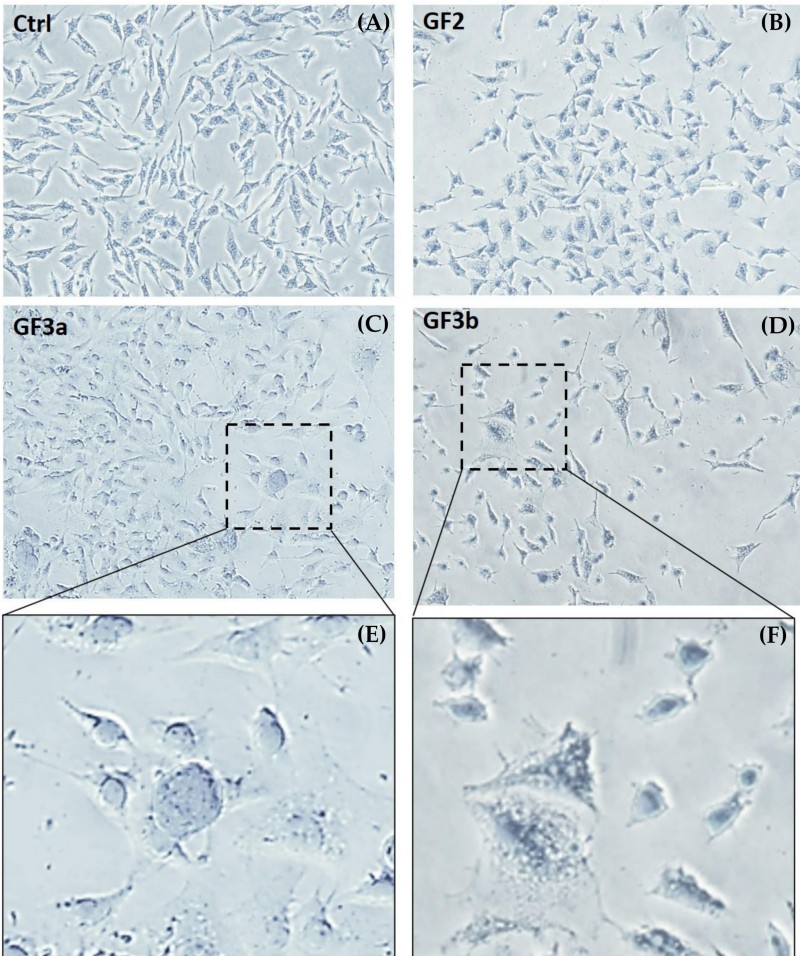

**Figure 4.** Representative images of HT22 cells treated with GF2, GF3a, and GF3b fibrous zeolites. The images were acquired by an optical microscope with a 10× objective after the treatment with 1 μm of GF2, GF3a, and GF3b zeolites. (**A**) control, (**B**) erionite GF2, (**C**) mesolite GF3a, and (**D**) thomsonite GF3b. The region bounded by the box is an enlarged region for the GF3a mesolite (**E**) and GF3b thomsonite (**F**) samples that highlight the morphologic changes of treated-HT22 cells. Scale bar 200 μm.

For the Jurkat cell line, our results showed a time-dependent decrease of viable cells for both concentrations tested (0.1 and 1 μm), but we did not observe a dose-dependent effect of zeolites (Figure 5). In particular, all zeolites tested did not show a significant cytotoxic effect within the first 24 h of exposure. However, a drop decrease in cell survival was evident at 48 h of treatment, and it was particularly significant with the fibrous zeolites GF2 (** $p < 0.01$).

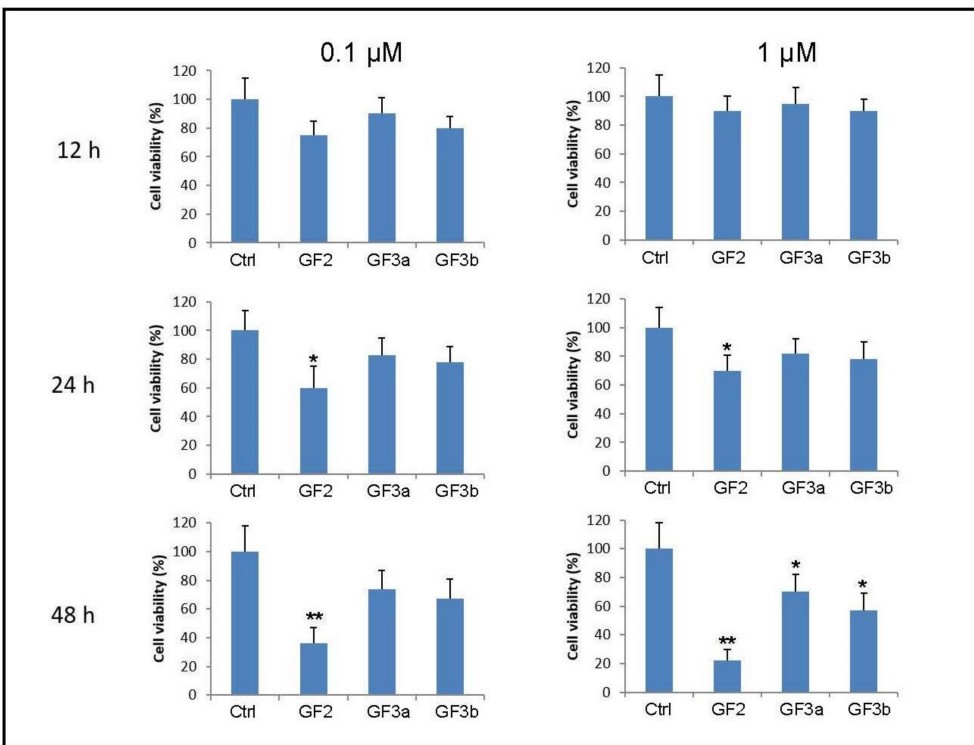

**Figure 5.** Evaluation of potential toxicity of GF2, GF3a, GF3b fibrous zeolites in Jurkat cell line. The histograms show cell viability after 12, 24, and 48 h of GF2, GF3a, and GF3b exposure at the concentration of 0.1 μM and 1 μM. Each value is expressed as a percentage ± SD (N = 3 independent experiments performed in triplicate; * $p < 0.05$, ** $p < 0.01$ vs. Ctrl).

## 4. Discussion and Conclusions

Naturally occurring zeolites are crystalline aluminosilicate materials with a three-dimensional structure containing cavities that can hold cations (e.g., Na, K, Ca, Mg) and water molecules [48–50]. The inhalation of these particles, especially those with a fibrous habit, can be responsible for local cellular inflammation, cytokine responses, silicosis, and increased lung cancer rates. Despite several studies in this field, the pathway of toxicity is still poorly understood [36,51,52], even if the role of mineral fibers in pulmonary fibrosis [53] and carcinogenesis [54] is increasingly noted. It has been demonstrated that the exposure of erionite fibers to humans is clearly related to malignant mesothelioma, and in vivo studies showed that erionite can be considered the most carcinogenic mineral [35,55]. Accordingly, growing attention focuses on other similar fibrous zeolites and their relations to environmental and occupational exposures [2,13,16,21–23,26,28–33].

It has already been shown that some zeolite fibers interact well with model membranes [33] and can interact with the human monocyte U937 cell line [34]. In particular, the zeolite fibers demonstrate different abilities to interact with the external cell surface, enter the cell membrane, reach the nucleus of the cells, and generate cell necrosis. Therefore, it becomes very interesting to see and try to understand how these fibrous zeolites interact with other cell lines and how fibers affect cells. This study investigated the potential toxicological implications of three natural fibrous zeolites in two different in vitro cellular models, the adherent murine hippocampal cell line (HT22) and the human immortalized T lymphocyte cell line (Jurkat). In particular, we studied an already known carcinogenic zeolite (GF2, erionite), a suspected carcinogenic mesolite (GF3a), and a supposed non-carcinogenic thomsonite (GF3b).

Our results demonstrated that the GF2 erionite, characterized by inhalable fibrils, showed cytotoxic effects on both HT22 and Jurkat cell lines. The GF2 sample has the highest specific surface area (8.14 m$^2$/g) and a homogeneous site distribution, as testified

by EPR measurements [32]. The significant interaction capability surface of the GF2 and strength of its interaction with cell membrane could be correlated to its cellular toxicity.

The GF3a mesolite sample mainly consists of tiny fibers and fibrils of inhalable size and rarely of prismatic crystals. Our data demonstrate that GF3a induces significant toxicity on the HT22 cell line while a weak response at high doses and after prolonged exposure in Jurkat cells. GF3a shows a smaller specific surface area ($1.55 \, \mathrm{m^2/g}$) and the surface is less homogeneous than GF2, showing the presence of close polar sites [32]. Here, the surface of the cell membranes may interact with the polar site on mesolite crystals and provoke a cytotoxic response in hippocampal HT22 cells. However, given the different cytotoxicity between the two cell lines, further in-depth studies on the interaction mechanisms of this zeolite are necessary.

Likewise, the GF3b thomsonite can induce a toxic response only at the high concentration tested in both cell lines. The GF3b thomsonite consists of massive prismatic crystals and lamellae of bigger size, which are then not inhalable. GF3b sample has the lowest specific surface area ($0.38 \, \mathrm{m^2/g}$). Similar to mesolite, thomsonite shows a less homogeneous surface than GF2 and the presence of close polar sites [32]. In this case, the superficial characteristics may also be correlated with an easier interaction of GF3b with the cell membrane. However, the effects shown in the two cell lines suggest that GF3b thomsonite is less toxic than GF3a mesolite, but it cannot be considered a safe zeolite.

Considering these findings, this in vitro study is a starting point to screen minerals' capability to elicit cytotoxicity and cell damage in two different cellular models. Furthermore, the three fibers' different morphological and chemical–physical properties are the purpose for better understanding of the potential mechanisms of action underlying the other toxic behaviors.

In conclusion, this work gives novel insights into the knowledge of the toxicological implications of erionite and the newly characterized zeolites mesolite and thomsonite. Obviously, for a complete understanding of the mechanisms of the cellular/tissue responses to these fibrous zeolites, in vivo animal tests should be performed and compared to the outcome of the in vitro tests.

**Author Contributions:** Conceptualization, methodology, data curation, formal analysis, M.B.; data curation, formal analysis, investigations, writing-original draft preparation, M.G.N., F.L., and M.G.; writing-review and editing, supervision, project administration, funding acquisition, M.M. All authors have read and agreed to the published version of the manuscript.

**Funding:** This research was conducted under the "Fibers a Multidisciplinary Mineralogical, Crystal-Chemical and Biological Project to Amend the Paradigm of Toxicity and Cancerogenicity of Mineral Fibers" (PRIN: PROGETTI DI RICERCA DI RILEVANTE INTERESSE NAZIONALE-Bando, 2017-Prot. 20173X8WA4). This work was also funded under the framework of the 2020 research programs of the Department of Pure and Applied Sciences of the University of Urbino Carlo Bo (project "New asbestiform fibers: mineralogical and physical-chemical characterization", responsible, M. Mattioli) and the INAIL-BRIC ID60 Project (responsible, M. Mattioli).

**Data Availability Statement:** Not applicable.

**Acknowledgments:** We warmly thank Laura Valentini for her kind assistance during the SEM-EDS analyses. We are also thankful to Dodie James for her help in revising the English style. Very constructive suggestions by three anonymous reviewers significantly improved the manuscript.

**Conflicts of Interest:** The authors declare no conflict of interest. The funders had no role in the design of the study; in the collection, analyses, or interpretation of data; in the writing of the manuscript, or in the decision to publish the results.

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
