# Peer review of "Potential Toxicity of Natural Fibrous Zeolites: In Vitro Study Using Jurkat and HT22 Cell Lines"

_minerals, doi:10.3390/min12080988_

Round 1

Reviewer 1 Report

In general, this paper provides some interesting data concerning the shape and chemical composition of different fibrous zeolites. The authors also provide basic data suggesting cytotoxicity of these fibers. However, overall, there is not enough novel information to warrant publication. If the authors included additional toxicology assays and better explained the rationale behind the choice of experimental systems, this paper would be more interesting and provide more novel and relevant information to the field. Specific suggestions are below.

Major comments

1.       The in vitro dosing needs more explanation. It is common to have fiber dosage presented as µg/cm^2, not as molarity.

2.       How were the fibers prepared for cell treatment? Were they elutriated or were raw fibers used? Any size selection? Were solutions sonicated or just vortexed prior to dosing? What solution were the fibers prepared in?

3.       Choice of cell line needs to be justified. Authors chose mouse hippocampal and human T cell; neither of these cells would be considered “first responders” to inhaled fibers nor would they be site of initial toxicity. Therefore, justification of choice needs to be explicit.

4.       Choice of cytotoxicity assay should also be explained. While trypan blue is acceptable for detecting living cells, it is not the most accurate nor the most sensitive assay available. DNA-binding dyes are more commonly used for cytotoxicity assays. At the least, authors should consider adding additional assays to corroborate findings.

5.       Data and methods for Figure 3 needs more explanation. The authors state they performed clonogenic assays, but they microscope pictures don’t support this. The images show cell monolayers, not colonies. The authors also focus on cell phenotypes, which are important, but, again, do not demonstrate colony formation. Clonogenic assays are also performed after 1-3 weeks of incubation, not 48hrs (the methods state 1 week but the figure shows 48hrs). If the authors are wanting to show cell toxicity, as with the Jurkat cells, I would suggest an MTT or DNA-binding dye assay, which can be done on adherent cells. Alternatively, clarify the methods and results sections so they match and use microscope pictures that show colony formation in addition to the cell phenotypes.

6.       Additional cell assays would add novelty to this study. ROS and LDH assay data would be of interest and assays are simple to perform.

Minor issues

1.       Confirm journal preference in use of commas vs. decimals in numbers (0,1 vs 0.1M)

2.       Also confirm formatting of cell numbers (methods section, lines 102 & 108; 2x105 vs 2x10^5)

3.       Line 33, phrase “there is several evidence of” is confusing

4.       In vitro and in vivo studies of erionite toxicity have been performed by Zebedeo et al. but is not cited

5.       Lines 53-56 suggest that this study is looking at carcinogenicity of fibers, which it is not.

6.       Where did you get the cells from?

7.       Line 111-112, what is meant by cells were cultured for seven days depending on proliferation rate? Weren’t all cells cultured for the same period of time? If not, these data are not comparable between samples.

8.       Suggestion to label figure 1 panels as A, B, and C; this would help readers know which panel is referred to in the text.

9.       Figure 1 legend should specify which fiber samples correspond to GF2, GF3a, GF3b, instead of making reader refer to methods.

10.   Figure 3 legend, these are not histograms.

Author Response

Reviewer #1

In general, this paper provides some interesting data concerning the shape and chemical composition of different fibrous zeolites. The authors also provide basic data suggesting cytotoxicity of these fibers. However, overall, there is not enough novel information to warrant publication. If the authors included additional toxicology assays and better explained the rationale behind the choice of experimental systems, this paper would be more interesting and provide more novel and relevant information to the field. Specific suggestions are below.

ANSWER: we thank Reviewer#1 for our work's positive comments and constructive remarks.

First, we would like to point out that all the observations made by reviewer 1 are pertinent and highly relevant, and we have answered everything possible. However, it was impossible to satisfy some requests fully (e.g., part of points 4 and 6) for the following reason. This manuscript is directed to a special volume of Minerals on mineral fibers, and its primary connotation is mainly mineralogical, and non toxicological. Our goal, also according to the suggestion of the Editor, was to open a window on the possible interactions of fibrous minerals (never tested before) with some cell lines to provide a starting point for further research in this area and subsequent works of a more purely biological-toxicological type, certainly much complete and more structured than this manuscript. Including additional toxicological assays (e.g., ROS and LDH assay data), as suggested by reviewer#1, would certainly be more interesting and will be planned in future works.

Major comments

  1. The in vitro dosing needs more explanation. It is common to have fiber dosage presented as µg/cm^2, not molarity.

ANSWER: We agree with the reviewer’s comment. The authors chose to present the dosage of the fibers in molarity and not in µg/cm^2 considering this measure of dosage more correct since the cells grow in adhesion (HT22) and suspension (Jurkat).

  1. How were the fibers prepared for cell treatment? Were they elutriated or were raw fibers used? Any size selection? Were solutions sonicated or just vortexed prior to dosing? What solution were the fibers prepared in?

ANSWER: Raw fibers of GF2, GF3a and GF3b were selected from each sample by hand-picking working under a binocular optical microscope and were subsequently disaggregated and gently ground in an agate mortar, without any size selection. The fibers were resuspended in physiological solution, sonicated for 45 s at 100 W and immediately incubated with the cells. This last sentence has also been added to the text (Cell culture and treatment: see lines 93-94 of the revised manuscript - untracked version).

  1. Choice of cell line needs to be justified. Authors chose mouse hippocampal and human T cell; neither of these cells would be considered “first responders” to inhaled fibers nor would they be site of initial toxicity. Therefore, justification of choice needs to be explicit.

ANSWER: We agree with the reviewer on his/her comment about cell lines. We would like to underline that this is not exclusively a biological paper but rather lay the basis for a future biological project on inflammatory processes. In light of this, we have investigated two cell lines largely used in our laboratory, one related to the hematopoietic system and the other to the CNS in order to study the inflammatory effect of these compounds in different kinds of tissues.

  1. Choice of cytotoxicity assay should also be explained. While trypan blue is acceptable for detecting living cells, it is not the most accurateor sensitive assaye. DNA-binding dyes are more commonly used for cytotoxicity assays. At the least, authors should consider adding additional assays to corroborate findings.

ANSWER: We agree with the reviewer’s comment. The trypan blue exclusion assay allows a direct identification and enumeration of live (unstained) and dead (blue) cells in a given population. In this first paper, we needed to identify the doses and the times of incubation in order to define a cytotoxicity dose and a range of incubation times. In our opinion, this is particularly relevant for GF3a and GF3b crystals as there are no indications in the literature to date. For further investigations, we’ll perform other specific analyses for cell viability.

  1. Data and methods for Figure 3 needs more explanation. The authors state they performed clonogenic assays, but they microscope pictures don’t support this. The images show cell monolayers, not colonies. The authors also focus on cell phenotypes, which are important, but, again, do not demonstrate colony formation. Clonogenic assays are also performed after 1-3 weeks of incubation, not 48hrs (the methods state 1 week but the figure shows 48hrs). If the authors are wanting to show cell toxicity, as with the Jurkat cells, I would suggest an MTT or DNA-binding dye assay, which can be done on adherent cells. Alternatively, clarify the methods and results sections so they match and use microscope pictures that show colony formation in addition to the cell phenotypes.

ANSWER: We thank the reviewer for the suggestions. The data and methods in Figure 3 have been better described as requested by the reviewer. We also added the representative images of colony formation in Figure 3B. Regarding the clonogenic assay procedure, the test was performed after 1 week, but the cells were previously treated for 12, 24, or 48 hours.

  1. Additional cell assays would add novelty to this study. ROS and LDH assay data would be of interest and assays are simple to perform.

ANSWER: Because the pathways activated by these compounds are not fully elucidated, especially for the GF3a and GF3b, we are aware that the evaluation of ROS and LDH can give important suggestions. However, our intent was to deepen these biological aspects in subsequent research.

Minor issues

  1. Confirm journal preference in use of commas vs. decimals in numbers (0,1 vs 0.1M)

ANSWER: Yes, numbers have been modified according to the Journal guidelines.

  1. Also confirm formatting of cell numbers (methods section, lines 102 & 108; 2x105 vs 2x10^5)

ANSWER: Sorry for these inaccuracies; the cell numbers have been modified.

  1. Line 33, phrase “there is several evidence of” is confusing

ANSWER: This sentence has been modified according to the suggestion.

  1. In vitro and in vivo studies of erionite toxicity have been performed by Zebedeo et al. but is not cited

ANSWER: The reviewer is correct; this relevant paper has been cited as a reference [19].

  1. Lines 53-56 suggest that this study is looking at carcinogenicity of fibers, which it is not.

ANSWER: We agree with the reviewer’s comment; the sentence has been changed.

  1. Where did you get the cells from?

ANSWER: The information requested by the reviewer has been added to the revised manuscript.

  1. Line 111-112, what is meant by cells were cultured for seven days depending on proliferation rate? Weren’t all cells cultured for the same period of time? If not, these data are not comparable between samples.

ANSWER: HT22 was cultured for seven days for all samples tested. The sentence means that HT22 has a rapid proliferative rate (16 hours) and is needed only 1 week for optimal colony formation. Anyways, the sentence has been changed.

  1. Suggestion to label figure 1 panels as A, B, and C; this would help readers know which panel is referred to in the text.
  2. Figure 1 legend should specify which fiber samples correspond to GF2, GF3a, GF3b, instead of making reader refer to methods.

ANSWER: We thank reviewer#1 for these suggestions. Figure 1 and its legend have been modified accordingly.

  1. Figure 3 legend, these are not histograms.

ANSWER: The legend of Figure 3 has been corrected.

Reviewer 2 Report

In this manuscript, the authors present an initial study to investigate the toxic effects of natural fibrous zeolites (erionite, mesolite, and thomsonite) via characterization and cellular tests. The results presented are promising and a good starting point for further research in this area. This opens up opportunities for research in this field important for human health.

Author Response

Reviewer #2

In this manuscript, the authors present an initial study to investigate the toxic effects of natural fibrous zeolites (erionite, mesolite, and thomsonite) via characterization and cellular tests. The results presented are promising and a good starting point for further research in this area. This opens up opportunities for research in this field important for human health.

ANSWER: thanks a lot to Reviewer#2 for her/his very positive comment on our work.

Reviewer 3 Report

This study provides novel insights into the knowledge of the toxicological implications of the erionite and the newly characterized zeolites mesolite and thomsonite. Overall, it is well written, concise, and schematic, effective in its discussion of data, with good English. Undoubtedly, the authors demonstrate a good grasp of the subject matter.

·        The introduction, although concise, provides the basic information with clarity and comprehensiveness, likewise the chapter "Materials and Methods".

·        Please check decimals (with a period) in the text, the abstract and the figures.

·        In Figure 1, the unit of measurement in abscissa is keV

·        In Figure 1 and 2 enter the unit of measurement in y (intensity, arbitrary unit)

·        In Figure 4, it would perhaps be useful to include a graphical scale.

·        I wonder if it is possible to define or compare the parameters of FPTI (fiber potential toxicity/pathogenicity index) as proposed by Gualtieri (2018).

Author Response

Reviewer #3

This study provides novel insights into the knowledge of the toxicological implications of the erionite and the newly characterized zeolites mesolite and thomsonite. Overall, it is well written, concise, and schematic, effective in its discussion of data, with good English. Undoubtedly, the authors demonstrate a good grasp of the subject matter. The introduction, although concise, provides the basic information with clarity and comprehensiveness, likewise the chapter "Materials and Methods".

ANSWER: we thank Reviewer#3 for the good comments about our work.

Please check decimals (with a period) in the text, the abstract and the figures.

ANSWER: Yes, all decimals have been checked and corrected.

In Figure 1, the unit of measurement in abscissa is keV

ANSWER: Yes, sorry for this inaccuracy. Units have been corrected.

In Figure 1 and 2 enter the unit of measurement in y (intensity, arbitrary unit)

ANSWER: Ok, done.

In Figure 4, it would perhaps be useful to include a graphical scale.

ANSWER: As suggested by the referee, scale bars were added in figure 4.

I wonder if it is possible to define or compare the parameters of FPTI (fiber potential toxicity/pathogenicity index) as proposed by Gualtieri (2018).

ANSWER: Yes, we agree with reviewer#3. It should be of great interest to describe and compare, for the investigated minerals, all the parameters defined in the FPTI Model of Gualtieri (2018). However, such comparison will be possible only after a complete characterization of these fibers, for which additional investigations are needed.